# Semaphorin 6D–expressing mesenchymal cells regulate IL-10 production by ILC2s in the lung

Maiko Naito[1,2], Yoshimitsu Nakanishi[1,2,3], Yasutaka Motomura[4,5,6], Hyota Takamatsu[1,2], Shohei Koyama[1,2,7], Masayuki Nishide[1,2], Yujiro Naito[1,2], Mayuko Izumi[1,2], Yumiko Mizuno[1,2], Yuta Yamaguchi[1,2], Satoshi Nojima[2,8], Daisuke Okuzaki[3,9,10], Atsushi Kumanogoh[1,2,3,10]

**Group 2 innate lymphoid cells (ILC2s) have been implicated in both physiologic tissue remodeling and allergic pathology, yet the niche signaling required for ILC2 properties is poorly understood. Here, we show that an axonal guidance cue semaphorin 6D (Sema6D) plays critical roles in the maintenance of IL-10–producing ILC2s. *Sema6d*[−/−] mice exhibit a severe steady-state reduction in ILC2s in peripheral sites such as the lung, visceral adipose tissue, and mesentery. Interestingly, loss of Sema6D results in suppressed alarmin-driven type 2 cytokine production but increased IL-10 production by lung ILC2s both in vitro and in vivo. Consequently, *Sema6d*[−/−] mice are resistant to the development of allergic lung inflammation. We further found that lung mesenchymal cells highly express Sema6D, and that niche-derived Sema6D is responsible for these phenotypes through plexin A1. Collectively, these findings suggest that niche-derived Sema6D is implicated in physiological and pathological characteristics of ILC2s.**

## Introduction

Group 2 innate lymphoid cells (ILC2s) are a subset of innate lymphoid cells that functionally mirror CD4[+] T helper type 2 cells (Th2). In contrast to the antigen-specific response of Th2 cells, the activation of ILC2s is independent of antigen stimulation (Vivier et al, 2018). ILC2s produce type 2 cytokines in response to non-specific alarmins such as IL-33, IL-25, and thymic stromal lymphopoietin (Moro et al, 2010; Neill et al, 2010; Price et al, 2010). ILC2s are tissue-resident cells located in a variety of peripheral tissues (Gasteiger et al, 2015). Although ILC2s are important for inflammation, tissue remodeling, metabolism, and thermal homeostasis (McKenzie et al, 2014; Artis & Spits, 2015; Lee et al, 2015), these functions depend on the tissues in which they reside and on specific pathological conditions (Ricardo-Gonzalez et al, 2018; Schneider et al, 2019). Notably, lung ILC2s play a critical role in promoting allergic airway inflammation during innate immune responses (Halim et al, 2014; Martinez-Gonzalez et al, 2015). ILC2s produce IL-10 in addition to type 2 cytokines such as IL-4, IL-5, IL-9, and IL-13 (Seehus et al, 2017). IL-10 is an important multifunctional, anti-inflammatory cytokine that inhibits the outbreak of inflammatory cytokines, preventing host damage and maintaining the integrity of tissue function. The production of IL-10 is induced by cytokines (IL-33, IL-2, IL-4, and IL-27), retinoic acid (Morita et al, 2019), and neuropeptide neuromedin U (Bando et al, 2020). Recently, it is emerging that IL-10–producing ILC2s play important roles in human allergic disease. For instance, it has been reported that allergen-specific immunotherapy restored the ability of ILC2s to produce IL-10 (Boonpiyathad et al, 2021; Golebski et al, 2021), showing a potential of being new target for the treatment of allergic airway inflammation. However, it remains unclear how the signals within specific tissue environments help dictate the phenotype of regulatory ILC2s.

Semaphorins were originally identified as axon guidance factors during neuronal development (Kolodkin et al, 1993; Pasterkamp & Kolodkin, 2003). Cumulative findings have demonstrated that they have various effects in angiogenesis, tumor growth, bone homeostasis, and immune responses (Suzuki et al, 2007; Kang & Kumanogoh, 2013; Kumanogoh & Kikutani, 2013). Semaphorin 6D (Sema6D) is a class VI transmembrane-type semaphorin that functions as a ligand and a receptor through association with plexin A1. For instance, the Sema6D "reverse signal," acting as a receptor, induces PPARγ and leads to the reprogramming of lipid metabolism

[1]Department of Respiratory Medicine and Clinical Immunology, Graduate School of Medicine, Osaka University, Suita, Japan   [2]Department of Immunopathology, World Premier International Research Center Initiative (WPI), Immunology Frontier Research Center (IFReC), Osaka University, Suita, Japan   [3]Integrated Frontier Research for Medical Science Division, Institute for Open and Transdisciplinary Research Initiatives (OTRI), Osaka University, Suita, Japan   [4]Laboratory for Innate Immune Systems, Department for Microbiology and Immunology, Graduate School of Medicine, Osaka University, Suita, Japan   [5]Laboratory for Innate Immune Systems, WPI, Immunology Frontier Research Center (IFReC), Osaka University, Suita, Japan   [6]Laboratory for Innate Immune Systems, RIKEN Center for Integrative Medical Sciences (IMS), Yokohama, Japan   [7]Division of Cancer Immunology, Research Institute/Exploratory Oncology Research and Clinical Trial Center (EPOC), National Cancer Center, Chiba, Japan   [8]Department of Pathology, Graduate School of Medicine, Osaka University, Suita, Japan   [9]Genome Information Research Center, Research Institute for Microbial Diseases, Osaka University, Suita, Japan   [10]Center for Infectious Diseases for Education and Research (CiDER), Osaka University, Suita, Japan

Correspondence: kumanogo@imed3.med.osaka-u.ac.jp

in the context of macrophage polarization (Kang et al, 2018). On the other hand, the Sema6D "forward signal," acting as a ligand, regulates T-cell activation during the late phases of an immune response (O'Connor et al, 2008). However, the involvement of Sema6D in other immune cell functions remains unclear.

Here we demonstrated that lung mesenchymal cells expressing Sema6D is relevant to the ability to produce IL-10 by ILC2s. Our findings indicate that Sema6D signals in the lung tissue niche play critical roles in controlling regulatory functions of ILC2s.

# Results

## Loss of Sema6D reduces ILC2s in peripheral tissues

To investigate the role of Sema6D in mature ILC2s, we examined the populations of ILC2s in Sema6D-deficient (*Sema6d$^{-/-}$*) mice. Flow cytometric analysis showed that under steady-state conditions, Sema6D deficiency reduced the numbers of ILC2s in the lung, visceral adipose tissue (VAT), and mesentery, but not in the BM (Fig 1A and B).

We next analyzed cell surface marker expression on ILC2s to determine whether Sema6D deficiency affected basal ILC2 status. Lung ILC2s from *Sema6d$^{-/-}$* mice showed increased KLRG1 and PD-1 expression, both of which are known activation markers of ILC2s (Hoyler et al, 2012). In addition, *Sema6d$^{-/-}$* ILC2s exhibited downregulation of Thy1.2 expression (Fig 1C and D). Elevated expression of KLRG1 was also seen in VAT and mesentery ILC2s, but not in BM ILC2s (Fig 1E). These results suggest that Sema6D affects the activation status of ILC2s in peripheral tissues under steady-state conditions.

To assess the role of Sema6D in the development of ILC2s, we analyzed the expression of their related transcriptional factors and the numbers of common lymphoid progenitors (CLPs), α-lymphoid progenitors (α-LPs), common progenitor to all helper-like ILCs (CHILPs), and ILC2 progenitors (ILC2Ps) (Yu et al, 2014; Antignano et al, 2016; Monticelli et al, 2016). *Sema6d$^{-/-}$* and wild-type (WT) mice showed comparable numbers of ILC2 progenitors (Fig 1F), and Sema6D deficiency did not affect the expression of transcriptional regulators *Id2* and *Gata3*, which mark the ILC2 lineage (Fig 1G). Moreover, ILC2 progenitors expressed lower levels of *Sema6d* and its receptor *Plxna1* compared with mature ILC2s (Fig 1H). These data indicate that Sema6D deficiency does not affect the development of ILC2 progenitors in the BM but does contribute to maintaining the number of ILC2s in peripheral tissues.

## Sema6D deficiency impairs ILC2-induced type 2 inflammation

We next examined ILC2 effector functions after their activation. First, we cultured ILC2s from the lungs of *Sema6d$^{-/-}$* and WT mice with IL-2 and IL-33, the combination of which is the most potent known ILC2 stimulator and is critical for establishing allergic inflammation. ILC2s from *Sema6d$^{-/-}$* mice exhibited decreased IL-5 and IL-13 but increased IL-10 production in response to IL-2 and IL-33 in vitro (Fig 2A). The decreased expression of *Il5* and *Il13*, and elevated expression of *Il10*, were confirmed by qRT-PCR analysis

(Fig 2B). These data indicate that Sema6D deficiency suppresses type-2 cytokine production but promotes anti-inflammatory cytokine secretion by ILC2s.

To determine the pathological significance of Sema6D in allergic airway inflammation, we next performed ILC2-induced lung inflammation model by intranasal instillation of IL-33. *Sema6d$^{-/-}$* mice showed significantly reduced infiltration of eosinophils and ILC2s in bronchoalveolar lavage fluid (BALF) after IL-33 challenge (Fig 2C). Compared to the BALF of WT controls, that of *Sema6d$^{-/-}$* mice exhibited significantly decreased production of the type 2 cytokines IL-5 and IL-13, but increased production of IL-10 (Fig 2D). Consistent with these findings, *Sema6d$^{-/-}$* mice also demonstrated reduced infiltration of lung eosinophils, as well as a significantly lower number of ILC2s and a reduction in the capability of ILC2s to produce IL-5 and IL-13 (Fig 2E and F). However, *Il10* transcription levels were significantly increased in lung ILC2s in *Sema6d$^{-/-}$* mice (Fig 2G).

To confirm these results in a more physiologically relevant setting, *Alternaria alternata* (*A. alternata*) was used to induce lung inflammation (Maazi et al, 2015). Consistent with the observations from the IL-33 model, the numbers of eosinophils and ILC2s, the amount of IL-5 and IL-13 in BALF, the numbers of lung eosinophils and ILC2s, and the production of effector cytokines by ILC2s were dramatically decreased in *Sema6d$^{-/-}$* mice compared with WT mice (Fig 2H–K). Furthermore, *Sema6d$^{-/-}$* mice exhibited an increased amount of IL-10 in BALF (Fig 2I) and greater expression of the *Il10* gene (Fig 2L). The attenuation of lung inflammation was confirmed by histological analysis (Fig 2M). These results demonstrate that Sema6D is critical for ILC2-mediated type 2 lung inflammation.

## Sema6D suppresses the regulatory function of ILC2s

Focusing on IL-10 production by ILC2s, exhausted-like ILC2s, characterized by increased expression of T cell exhaustion markers, also produce large amounts of IL-10 (Miyamoto et al, 2019). To understand the mechanism of the increased production of IL-10 in ILC2s derived from *Sema6d$^{-/-}$* mice, we performed RNA sequencing analysis of *in vivo*-activated lung ILC2s from *Sema6d$^{-/-}$* and WT mice. In contrast with the increased expression of *Il10* in *Sema6d$^{-/-}$* ILC2s, the expression of T-cell exhaustion markers, including *Tigit* and *Ctla4*, was comparable between *Sema6d$^{-/-}$* and WT ILC2s. Moreover, the expression of *Lag3* was lower in *Sema6d$^{-/-}$* ILC2s. (Fig 3A). In addition, the expression of cytokines and their receptors was comparable between *Sema6d$^{-/-}$* and WT mice, with the exception of elevated expression of *Il10* in *Sema6d$^{-/-}$* mice (Fig 3B). Genes encoding transcriptional regulators associated with ILC2 development and/or function (*Id2* [Moro et al, 2010], *Gata3* [Hoyler et al, 2012], *Rora* [Wong et al, 2012; Halim et al, 2012], *Tcf7* [Yang et al, 2013], *Tox* [Seehus et al, 2015], *Bcl11b* [Walker et al, 2015], and *Gfi1* [Spooner et al, 2013]) were comparable between *Sema6d$^{-/-}$* and WT mice (Fig 3C). Of note, in comparison with WT ILC2s, *Sema6d$^{-/-}$* ILC2s showed significant increases in the expression of *Id3*, *Retnla*, and *Foxf1*, all of which are reported to be up-regulated in IL-10–producing ILC2s (Seehus et al, 2017) (Fig 3C). These data suggest that an up-regulated regulatory condition, rather than exhaustion, is seen in ILC2s derived from *Sema6d$^{-/-}$* mice.

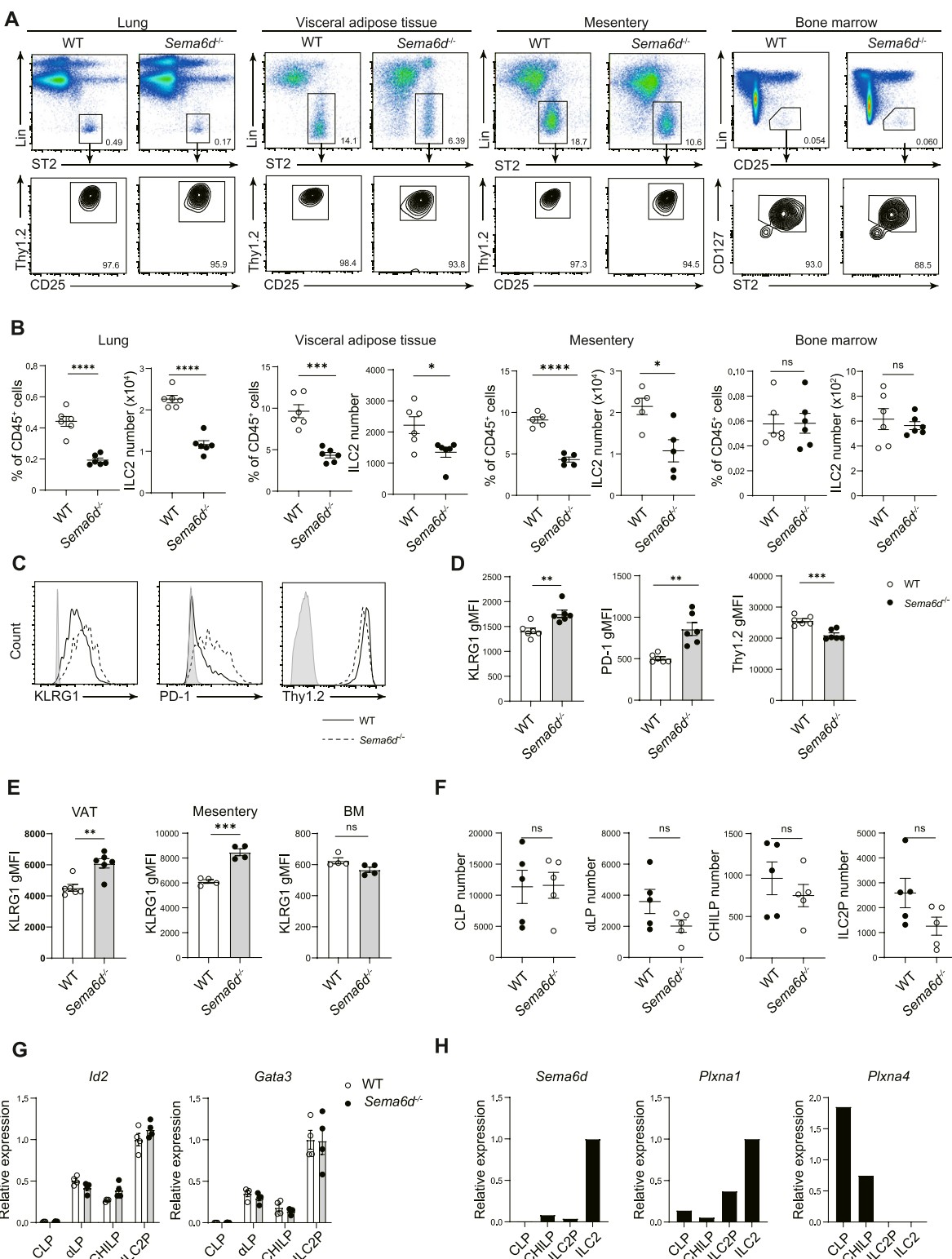

**Figure 1. Loss of Sema6D reduces ILC2s in peripheral tissues.**

**(A)** Representative flow cytometry plots of ILC2s in peripheral tissues and BM from WT and *Sema6d*$^{-/-}$ mice. **(B)** The frequencies and numbers of ILC2s (CD45$^+$ Lin$^-$ ST2+ CD90.2$^+$ Thy1.2$^+$) in (A) are shown (n = 4–6, per group). **(C)** Representative flow cytometric analysis of KLRG1, PD-1, and Thy1.2 expression, and geometric mean fluorescence intensity levels of lung ILC2s. **(D)** Geometric mean fluorescence intensity levels of KLRG1 in (A) are shown (n = 6, per group). **(E)** Geometric mean fluorescence intensity levels of KLRG1 in visceral adipose tissue, mesentery, and BM (n = 4–6, per group). **(F)** The numbers of CLP, α-LP, CHILP, and ILC2 cells in BM from WT and *Sema6d*$^{-/-}$ mice (n = 5, per group). **(G)** The mRNA expression of *Id2* and *Gata3* on BM CLP, α-LP, CHILP, and ILC2P cells measured by qRT-PCR. **(H)** The mRNA expression of *Sema6d*, *Plxna1*, and

To further confirm that deletion of Sema6D would not promote an exhausted-like phenomenon in ILC2s, we examined T-cell exhaustion marker expression on in vivo-activated lung ILC2s. Compared with lung ILC2s from WT mice, those from *Sema6d*$^{-/-}$ mice demonstrated greater expression of KLRG1 but comparable expression of PD-1, TIGIT, and CTLA4 (Fig 3D). Id3 is part of the inhibitors of differentiation (Id) protein family, which is a class of negative regulatory nuclear transcription factors (Yang et al, 2014). Id3 can positively regulate the differentiation of regulatory T (Treg) cells (Maruyama et al, 2011) and is also required for the development of IL-10–producing regulatory ILCs in the intestines (Wang et al, 2017). Moreover, high expression of Id3 was found in IL-10–producing ILC2s (Seehus et al, 2017), but not in exhausted-like ILC2s (Miyamoto et al, 2019). Lung ILC2s of *Sema6d*$^{-/-}$ mice stimulated with IL-33 and *A. alternata* showed increased Id3 expression (Fig 3E and F), indicating that regulatory function is up-regulated in *Sema6d*$^{-/-}$ ILC2s. In addition to gene expression profiles, proliferation capacity, which is impaired in exhausted-like ILC2s, was maintained in ILC2s derived from *Sema6d*$^{-/-}$ mice (Fig 3G and H). These data indicate that the exhausted-like phenomenon does not occur in *Sema6d*$^{-/-}$ lung ILC2s.

IL-10 exhibited an inhibitory effect on ILC2s and reduced type 2 cytokine production (Morita et al, 2015). We hypothesized that the reduction of type 2 cytokine production seen in *Sema6d*$^{-/-}$ ILC2s might be due to increased IL-10 production. Neutralization of IL-10 partially abrogated the decrease in IL-13 production by *Sema6d*$^{-/-}$ ILC2s in response to IL-33 and IL-2 (Fig 3I). In addition, qPCR analysis showed that IL-13 expression was increased by blocking IL-10 (Fig 3J). As expected, the low IL-13 production by ILC2s derived from *Sema6d*$^{-/-}$ mice resulted from increased IL-10 production. However, because neutralization of IL-10 alone does not fully restore IL-13 production, it remains possible that other pathways may contribute to the suppression of type 2 cytokine production in *Sema6d*$^{-/-}$ ILC2s. These results suggest that deletion of Sema6D does not induce an exhausted-like phenotype in activated ILC2s, but instead enhances the regulatory function of ILC2s by inducing IL-10 production, and partially contribute to the reduction of type 2 cytokines.

### Sema6D signaling from tissue niches suppresses IL-10–producing ILC2s

To investigate whether the ILC2 defects in *Sema6d*$^{-/-}$ mice had cell-intrinsic or -extrinsic causes, we generated *Sema6d*-deficient BM chimeric mice in hematopoietic or non-hematopoietic compartments. Mice lacking non-hematopoietic Sema6d (WT→*Sema6d*$^{-/-}$) exhibited fewer lung ILC2s and elevated KLRG1 expression compared with mice that systemically lacked Sema6d expression (*Sema6d*$^{-/-}$→*Sema6d*$^{-/-}$). Conversely, in mice lacking hematopoietic Sema6d (*Sema6d*$^{-/-}$→WT), lung ILC2 numbers and KLRG1 expression were comparable with those in WT mice (WT→WT; Fig 4A and B). These data indicate that the inability of non-hematopoietic cells to express Sema6D leads to a failure to sustain lung ILC2 number and activation status.

Next, we tested whether Sema6D was important in modulating ILC2 regulatory functions. ILC2s cultured in vitro with recombinant Sema6D showed decreased IL-10 production and *Il10* expression after stimulation with IL-2 and IL-33 (Fig 4C and D), although similar effects were not observed in control recombinant Sema3A proteins (Fig 4E). These results suggest that Sema6D down-regulates ILC2 regulatory functions.

Sema6D functions as a ligand through its receptors plexin A1 and plexin A4 (Kumanogoh & Kikutani, 2013). However, lung ILC2s hardly express *Plxna4* (Fig 4F), and moreover, ILC2 number and KLRG1 expression was comparable between plexin A4–deficient (*Plxna4*$^{-/-}$) mice and WT mice (data not shown), implying that the Sema6D-plexin A1 axis might regulate IL-10–producing ILC2s. To identify the receptor required for regulating IL-10 production of ILC2s, we generated ILC2s lacking plexin A1 or plexin A4 using lenti-CRISPRv2-GFP knockout vectors. In accordance with WT ILC2s, plexin A4–deficient ILC2s showed decreased IL-10 production when cultured with recombinant Sema6D, whereas plexin A1–deficient ILC2s were unaffected (Fig 4G). Collectively, these results confirm that plexin A1 is a functional receptor for Sema6D in regulating IL-10 production in ILC2s.

Given the importance of Sema6D in controlling ILC2 regulatory functions, we next determined its cellular source by isolating lung tissue. *Sema6d* mRNA was highly expressed in the CD45$^-$ fraction, and the most abundant *Sema6d*-expressing population was mesenchymal cells (Fig 4H). Lung ILC2s reside in the adventitial cuffs of lung vessels and airways, and are localized with a population of mesenchymal, adventitial stromal cells (Dahlgren et al, 2019). Lung mesenchymal cells are known to produce IL-33 to support ILC2s. However, the number of lung mesenchymal cells and their expression of *Il-33* mRNA were comparable in *Sema6d*$^{-/-}$and WT mice (Fig 4I). Moreover, co-culture of lung mesenchymal cells derived from *Sema6d*$^{-/-}$ mice with WT ILC2s resulted in increased IL-10 production compared with WT controls (Fig 4J). These data indicate that cell-extrinsic Sema6D signaling from lung mesenchymal cells controls the regulatory function of ILC2s.

## Discussion

In this study, we showed that Sema6D in lung niches suppresses the regulatory functions of ILC2s. ILC2s are long-lived, tissue-resident cells, and are infrequently replaced. The specific local microenvironmental niche seems to exert a dominant influence on ILC2 phenotypes, resulting in tissue-specific transcriptional signatures. Stromal cells acting as a reservoir for IL-33 have been identified in multiple tissues, such as lung, adipose tissue, liver, and brain meninges (Dahlgren et al, 2019; Rana et al, 2019). These IL-33–expressing niche cells all sustain a type 2 immune environment by inducing activation of ILC2s. In addition, IL-33, IL-2, and IL-4 are known to induce IL-10 production by ILC2s. Because these factors exacerbate type 2 inflammation, their anti-inflammatory effect is contradictory. In this context, Sema6D signaling by niche cells is remarkable because it may regulate the balance of pathogenic and

---

*Plxna*4 on BM CLP, CHILP, ILC2P, and ILC2 cells measured by qRT-PCR. Data are representative of three independent experiments (mean ± SEM). *$P$ < 0.05; **$P$ < 0.01; ***$P$ < 0.001 by $t$ test.

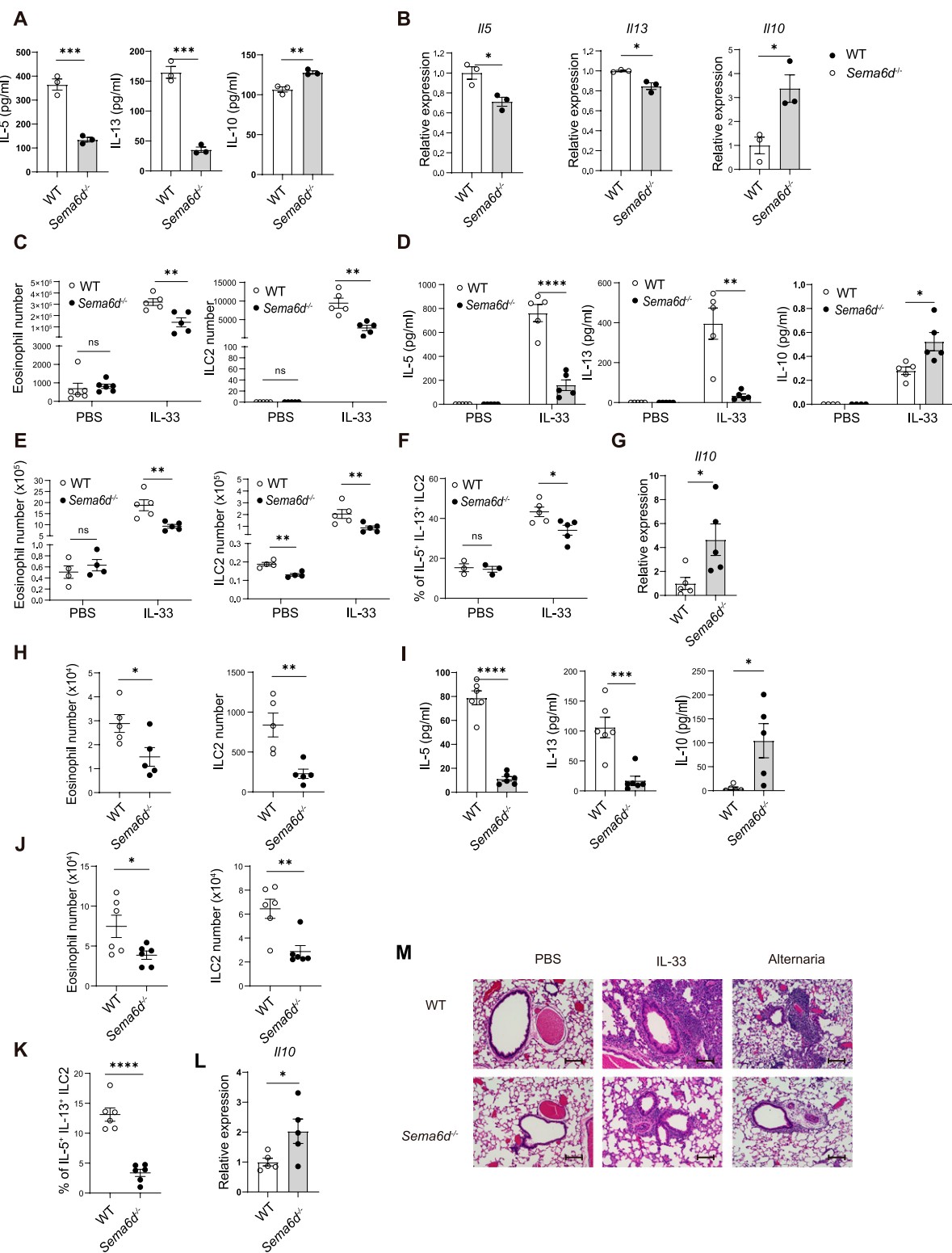

**Figure 2. Sema6D deficiency impairs type 2 cytokine production by ILC2 and attenuates ILC2-induced type 2 inflammation.**
**(A)** Equal numbers of lung ILC2s (1.5 × 10⁴) sorted from WT and *Sema6d*⁻/⁻ mice were cultured in the presence of IL-2 and IL-33 (10 ng/ml each) for 72 h. The amounts of IL-5, IL-13, and IL-10 in the culture supernatant were evaluated by ELISA. **(B)** The mRNA expression of *Il5*, *Il13*, and *Il10* in lung ILC2s was evaluated by qRT-PCR. **(C, D, E, F)** WT and *Sema6d*⁻/⁻ mice were intranasally administered IL-33 (500 ng/mouse/day) or PBS for 3 consecutive days (n = 5 or 6, per group). Mice were euthanized on day 4. **(C)** The numbers of eosinophils and ILC2s in BALF from WT and *Sema6d*⁻/⁻ mice after PBS or IL-33 treatment. **(D)** The amounts of IL-5, IL-13, and IL-10 in BALF were determined by ELISA and cytometric bead array. **(E)** The numbers of eosinophils and ILC2s in the lung after PBS or IL-33 treatment. **(F)** The frequencies of IL-5⁺ IL-13⁺ lung ILC2s after 3-h

regulatory functions of ILC2s by controlling IL-10 production, without activating them. Niche cells not only support the activation of ILC2s but also define their pathogenicity and regulatory functions. Under steady-state conditions, Sema6D-expressing lung niche cells might preserve the number and pathogenic features of ILC2s for the protection from inhaled allergens. Under inflammatory states, however, proliferated ILC2s, which lost contact with Sema6D-expressing cells, might up-regulate regulatory functions to prevent prolonged inflammation that could result in tissue damage such as airway remodeling. Concerning pathological condition of asthma, blockage of Sema6D signal from the tissue niches might have therapeutic potential of attenuating airway type 2 inflammation by up-regulating IL-10 production from ILC2s. In our study, the number of ILC2s in $Sema6d^{-/-}$ mice was lower not only in the lung, but also in the mesentery and VAT, implying that Sema6D-expressing stromal cells might act to control IL-10–producing ILC2 in these tissues as well. Further research is needed to demonstrate the molecular mechanisms of Sema6D in regulating ILC2s.

Regarding IL-10–producing properties, Sema6D has dual effects on immune cells. We have presented that Sema6D, acting as a ligand via its receptor plexin A1, suppressed IL-10 production by restricting the regulatory function of ILC2s. On the other hand, we have previously reported that, in macrophages, Sema6D acts as a receptor and promotes IL-10 production by inducing M2 polarization (Kang et al, 2018). Although further studies are required to define the molecular actions of Sema6D, its activities either as a ligand or receptor, may depend on physiological and pathological conditions. In addition to plexins, neuropilins have been identified as a primary semaphorin receptor. Plexin A1 serves as a direct binding receptor for class VI semaphorins, whereas class III semaphorins bind a receptor complex formed by plexin A1 and neuropilin-1 (Nrp1) (Kumanogoh & Kikutani, 2010). Although Sema3A-plexin A1/Nrp1 signaling was not involved in regulating IL-10 production by ILC2s in our experimental settings, TGFβ1–Nrp1 signaling was reported to enhance IL-5 and IL-13 production by ILC2s (Zhang et al, 2022).

In summary, our results reveal that Sema6D signaling in the lung tissue niche plays important roles in controlling the pathogenic and regulatory functions of ILC2. Defects in this pathway lead to increased regulatory function of ILC2s, resulting in attenuated type 2 inflammation. Our findings provide new insights into how the complicated crosstalk between the lung tissue niche and the immune system affect allergic inflammation in the lung.

# Materials and Methods

## Animals

C57BL/6J mice were obtained from CLEA Japan, Inc. $Sema6d^{-/-}$ mice were generated as previously described (Takamatsu et al, 2010).

B6.SJL-Ptprca Pepcb/BoyJ (CD45.1, #002014; Jax) mice were obtained from Jackson Laboratory. All mice were in the C57BL/6J background. 7- to 12-wk-old mice were used, and mouse experiments were randomized. For BM reconstitution, $1 × 10^7$ total BM cells were intravenously transferred to recipient mice that had been lethally irradiated with a single dose of 10 Gy. Mice were then used for experiments at least 8 wk after reconstitution. All mice used in this study were housed in a specific pathogen–free facility. All protocols were approved by the Animal Research Committee of the Immunology Frontier Research Center (Osaka University).

## Lung inflammation models

For IL-33–induced allergic inflammation, mice were intranasally administered 500 ng recombinant IL-33 (500 ng/mouse; R&D Systems) in 50 µl PBS for 3 consecutive days, and PBS was used as vehicle control. For *A. alternata* (*A. alternata*)–induced allergic inflammation, mice were intranasally administered *A. alternata* (30 µg/mouse, ITEA) in 30 µl PBS for 4 consecutive days. Mice were euthanized 24 h after the last challenge, and the lung and BALF were collected for analysis.

## Lung histology

Lung tissues were fixed in 4% PFA for 24 h and embedded in paraffin. Tissue sections were prepared and stained with hematoxylin–eosin (H&E) to evaluate inflammation.

## Isolation of cells from tissue

BM cells were obtained by flushing femurs and tibias with a syringe containing RPMI-1640 medium. Red blood cells were lysed with ammonium-chloride-potassium (ACK) buffer. VAT was removed and cut into small pieces with scissors and digested with Liberase TM (50 µg/ml; Roche) and DNase I (1 µg/ml; Roche) for 30 min at 37°C with continuous agitation in an incubator. The crude suspensions were further filtered through 70-µm cell strainers and the remaining red blood cells were lysed with ACK buffer. BALF was collected by flushing the lungs with 0.5 ml of cold PBS three times via a thin tube inserted into a cut made in the trachea, as previously described (Monticelli et al, 2016). A suspension of lung and mesentery cells were collected as previously described (Moro et al, 2015).

## Antibodies and reagents

mAbs specific for mouse CD3ε (145-2C11), CD4 (GK1.5), CD5 (53-7.3), CD8a (53-6.7), CD11c (HL3), CD16/CD32 (2.4G2), CD19 (1D3), CD25 (PC61), Gr-1 (RB6-8C5), NK1.1 (PK136), Siglec-F (E50-2440), ST2 (U29-

---

treatment with cell-stimulation cocktail. **(G)** The mRNA expression of *Il10* in lung ILC2s was evaluated by qRT-PCR. **(H, I, J, K, L)** WT and $Sema6d^{-/-}$ mice were intranasally administered *A. alternata* for 4 d (n = 5). Mice were euthanized 24 h after the last challenge. **(H)** The numbers of eosinophils and ILC2s in BALF were evaluated by flow cytometry. **(I)** The amounts of IL-5, IL-13, and IL-10 in BALF were measured by ELISA and cytometric bead array. **(J)** Total numbers of eosinophils and ILC2s in the lung after *A. alternata* treatment. **(K)** The frequencies of IL-5$^+$IL-13$^+$ lung ILC2s after 3-h treatment with cell-stimulation cocktail. **(L)** The mRNA expression of *Il10* in lung ILC2s was evaluated by qRT-PCR. **(M)** Representative H&E staining of lung sections in PBS, IL-33, and *A. alternata* groups (bars, 100 µm). Data are representative of two independent experiments (mean ± SEM). *$P < 0.05$, **$P < 0.01$, ***$P < 0.001$ by t test.

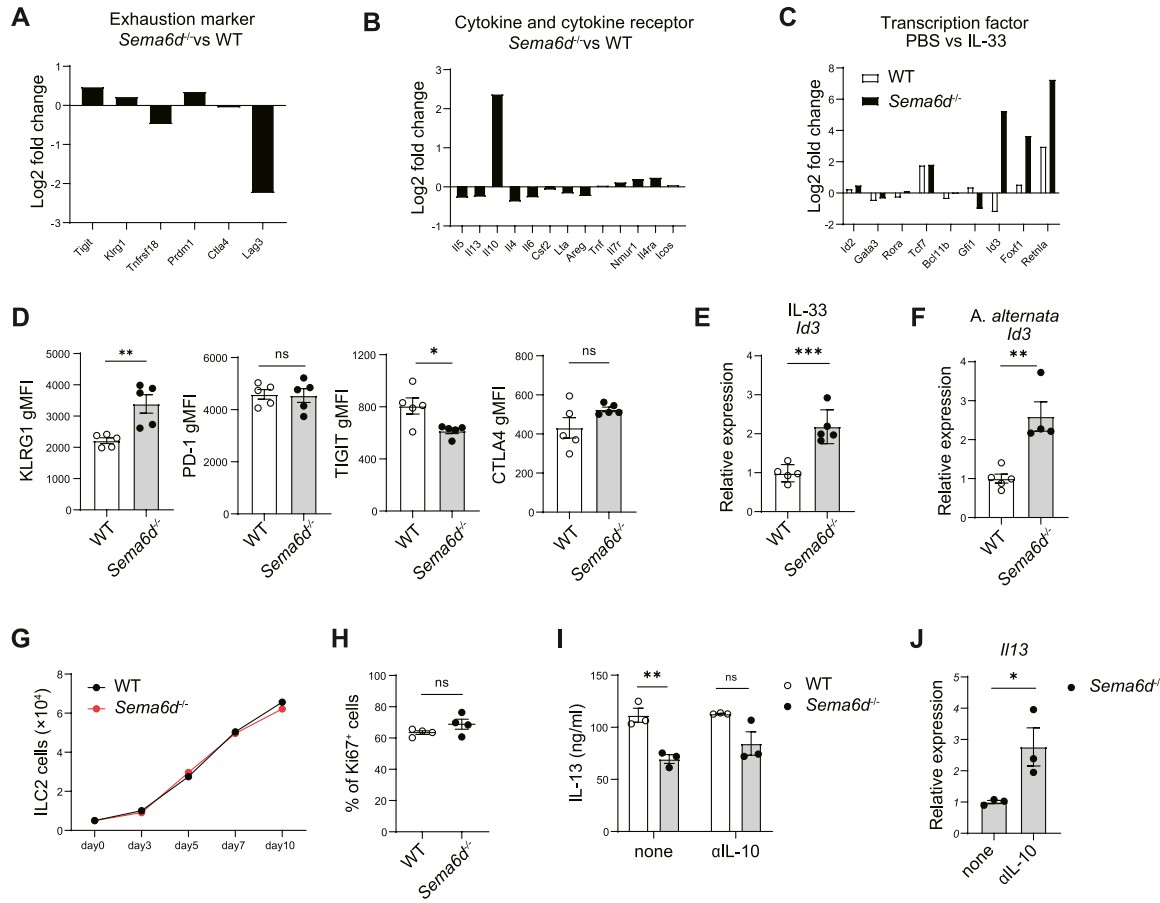

**Figure 3. Sema6D suppresses the regulatory features of ILC2s.**

**(A, B)** Mean fold changes of RPKM values of the indicated genes in ILC2s from $Sema6d^{-/-}$ mice versus WT mice after stimulation with IL-33 for 3 d. **(C)** Mean fold changes of RPKM values of the indicated genes in ILC2s stimulated with PBS versus IL-33. **(D)** WT mice and $Sema6d^{-/-}$ mice were treated with IL-33. The expression of the indicated genes in lung ILC2s was evaluated with qRT-PCR. These RNA samples were independent from those used to generate RNA-seq data. **(E, F)** WT and $Sema6d^{-/-}$ mice were treated with IL-33 or $A.\ alternata$. The expression of $Id3$ in lung ILC2s was evaluated with qRT-PCR. **(G)** Quantification of lung ILC2 cells ($5 \times 10^3$ cells; duplicate wells) cultured for 10 d with IL-2 and IL-33 (10 ng/ml each). **(H)** Frequencies of Ki67$^+$ ILC2s from WT and $Sema6d^{-/-}$ mice. **(I)** Concentrations of IL-13 in the supernatants of ILC2s ($1 \times 10^4$ cells) isolated from the lungs of WT and $Sema6d^{-/-}$ mice, cultured in the presence of IL-2 and IL-33 with or without anti-IL-10 for 72 h, as determined by ELISA. **(J)** The mRNA expression of $Il13$ in (I) was evaluated by qRT-PCR. Data are representative of three independent experiments (mean ± SEM). *$P < 0.05$; **$P < 0.01$; ***$P < 0.001$ by $t$ test. RPKM, reads per kilobase of exon per million mapped reads.

93), and Thy1.2 (53-2.1) were purchased from BD Biosciences. mAbs specific for mouse α4β7 (DATK32), CD11b (M1/70), CD31 (MEC13.3), CD45 (30F11), CD45.1 (A20), CD45.2 (104), CD127 (A7R34), EpCAM (G8.8), erythroid cell marker (TER-119; TER-119), FcγRIα (MAR-1), Flt3 (A2F10), Ki67 (16A8), PDGFRα (APA5), PD-1 (29F.1A12), Sca-1 (D7), TIGIT (1G9), IL-5 (TRFK5), and fluorochrome-conjugated streptavidin were purchased from BioLegend. mAbs specific for mouse CD25 (PC61.5), CTLA4 (UC10-4B9), IL-13 (eBio13A), KLRG1 (2F1), and F4/80 (BM8) were purchased from eBioscience.

Recombinant mouse IL-2, IL-7, IL-33, Sema3A-Fc, and Sema6D-Fc were purchased from R&D Systems. Anti-IL-10 mAb (JES5-2A5) was purchased from Bio X cell.

## Flow cytometric analysis and sorting

Mouse cells were blocked with anti-CD16/32 to block nonantigen-specific binding of Igs to Fcγ receptors. Live/Dead Fixable Dead Cell Stain Kit (Molecular Probes) was used to stain dead cells. For intracellular cytokine staining, the cells were stimulated with phorbol 12-myristate 13-acetate (PMA) (50 ng/ml; Sigma-Aldrich), ionomycin (1 mg/ml; Sigma-Aldrich), and brefeldin A (1 mg/ml; Thermo Fisher Scientific) for 3 h at 37°C, then fixed and permeabilized using the Intraprep Permeabilization Reagent (Beckman Coulter). Foxp3/Transcription Factor Staining Buffer Set (eBioscience) was used for staining CTLA4 and Ki67. mAbs against CD3ε, CD4, CD5, CD8α, CD11c, CD19, FcγRIα, F4/80, Gr-1, NK1.1, and TER-119 were used as Lin markers for the detection of ILC2s. mAbs against CD3ε, CD11b, CD19, F4/80, Gr-1, NK1.1, and TER-119 were used as Lin markers for the detection of CLP, α-lymphoid progenitor (αLP), common progenitor to all helper-like ILC (CHILP), and ILC2 progenitor (ILC2P). Lung, VAT, and mesentery ILC2s were gated by CD45$^+$ Lin$^-$ ST2$^+$ Thy1.2$^+$ CD25$^+$. BM ILC2s were gated by CD45$^+$ Lin$^-$ CD25$^+$ ST2$^+$ CD127$^+$. CLP were gated by CD45$^+$ Lin$^-$ CD127$^+$ α4β7$^-$ Flt3$^+$, α-LP were gated by CD45$^+$ Lin$^-$ CD127$^+$ α4β7$^+$ Flt3$^-$, ChILP were gated by CD45$^+$ Lin$^-$ CD127$^+$ α4β7$^+$ Flt3$^-$ CD25$^-$, and ILC2P were gated by CD45$^+$ Lin$^-$ CD127$^+$ α4β7$^+$ Flt3$^-$ CD25$^+$. Eosinophils were gated by CD45$^+$ Siglec-F$^+$

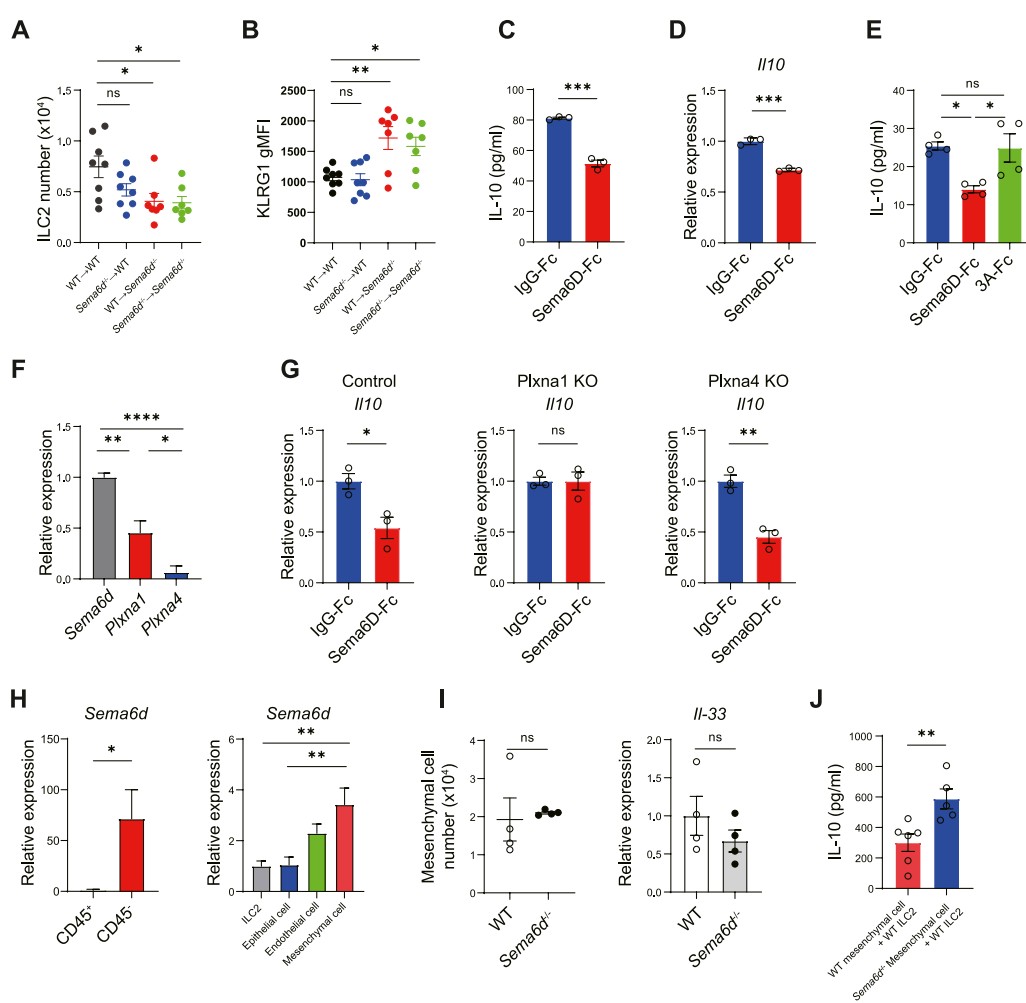

**Figure 4. Sema6D signaling from tissue niches suppresses IL-10–producing ILC2s.**
**(A, B)** Lung ILC2 numbers and KLRG1 expression in BM chimeric mice of the indicated genotypes. **(C, D, E)** Lung ILC2s from WT mice were cultured with Sema6D-Fc, Sema3A-Fc, or IgG-Fc (10 nM each) and stimulated with IL-2 and IL-33 (10 ng/ml each) for 3 d. **(C)** The amount of IL-10 in the supernatant was measured by ELISA. **(D)** The mRNA expression of *Il10* in (C) was evaluated by qRT-PCR. **(E)** The amount of IL-10 in the supernatant was measured by ELISA. **(F)** The mRNA expression of *Sema6d*, *Plxna1*, and *Plxna4* in WT lung ILC2s was evaluated by qRT-PCR. **(G)** Freshly sorted ILC2s were prestimulated with IL-2 and IL-33 on day 0 and infected with lentiviral vector (Plxna1 KO, Plxna4 KO, or control vector [mock]) on day 1 and 2. Puromycin-resistant cells were sorted as infected cells on day 7 and stimulated with IL-2 and IL-33 for 72 h. The mRNA expression of *Il10* in ILC2s was evaluated by qRT-PCR. **(H)** qRT-PCR analysis of the *Sema6d* mRNA in lung cells isolated from WT mice at steady state. **(I)** The lung mesenchymal cell number and *Il-33* mRNA expression were evaluated by qRT-PCR. **(J)** Concentrations of IL-10 in the supernatants of WT lung ILC2s after 3 d of co-culture with lung mesenchymal cells. Data are representative of two independent experiments (mean ± SEM). *$P < 0.05$; **$P < 0.01$; ***$P < 0.001$ by one-way ANOVA with Tukey–Kramer test (A, B, E, F, H) or $t$ test (C, D, E, G, H, I, J).

CD11b⁺ CD11c^low SSC^high. Epithelial cells were gated by CD45⁻ EpCAM⁺ CD31⁻. Endothelial cells were gated by CD45⁻ CD31⁺ EpCAM⁻. Mesenchymal cells were gated by CD45⁻ CD31⁻ EpCAM⁻ PDGFRα^dim+ Sca-1⁺. ILC2s as CD45⁺ Lin⁻ Thy1.2⁺ ST2⁺ were sorted from the lungs. Cells were analyzed using a FACSCanto II (BD Biosciences) and sorted using a FACSAria II (BD Biosciences). Data were analyzed using FlowJo software (BD Biosciences).

### In vitro ILC2 culture and cytokine quantification

For cytokine production analysis, ILC2s isolated from the lungs were seeded on 96-well round-bottom plates and stimulated in ILC2 culture medium supplemented with 10 ng/ml rmIL-2 and 10 ng/ml rmIL-33 at 37°C under 5% $CO_2$. The supernatants were used to determine the concentrations of IL-5, IL-13, and IL-10 using Quantikine ELISA Kits or DuoSet ELISA Kits (R&D Systems).

For mesenchymal cell–ILC2 co-culture, PDGFRa⁺ Sca1⁺ stromal cells were seeded on flat-bottomed 96-well plates in 200 μl DMEM (supplemented with 10% FBS, 50 U/ml penicillin, and 50 μg/ml streptomycin) at a density of 12,000 cells per well and allowed to form monolayers over 5–8 d. Sorted ILC2s were seeded onto the stromal cells at a density of 50,000–100,000 ILC2s per well, as previously described (Dahlgren et al, 2019).

### Quantitative real-time PCR (qRT-PCR)

Total RNA was isolated from sorted cells using RNAClean XP (Beckman Coulter), and cDNA was synthesized with SuperScript IV

Reverse Transcriptase (Invitrogen). Quantitative PCR reactions were established using QuantiFast Multiplex PCR Kits (QIAGEN) and run on QuantStudio 7 (Applied Biosystems). The following primers were used: GATA Binding Protein 3 (*Gata3*; Mm00484683), inhibitor of DNA binding 2 (*Id2*; Mm00711781_m1), inhibitor of DNA binding 3 (*Id3*; Mm00492575_m1), interleukin-5 (*Il5*; Mm00439646_m1), interleukin-10 (*Il10*; Mm00439616_m1), interleukin-13 (*Il13*; Mm00434204_m1), interleukin-33 (Il33; Mm00505403_m1), plexin A1 (*Plxna1*; Mm00501110_m1), plexin A4 (*Plxna4*; Mm01163292_m1), RAR-related orphan receptor A (*Rora*; Mm01173766_m1), semaphorin 6d (*Sema6d*; Mm00553142_m1), and endogenous control gene *Actb* (4352341E, Applied Biosystems). The gene expression data were normalized by the expression of an endogenous control gene.

### RNA sequencing

Library preparation was performed using a TruSeq Stranded mRNA Sample Prep Kit (Illumina). Sequencing was performed on an Illumina HiSeq 2500 platform (Illumina) in 75-base single-end mode. CASAVA 1.8.2 software (Illumina) was used for base calling. Sequenced reads were mapped to the mouse reference genome sequence (mm10) using TopHat v2.0.13 in combination with Bowtie2 ver. 2.2.3 and SAMtools ver. 0.1.19. Fragments per kilobase of exon per million mapped fragments (FPKMs) were calculated using Cuffnorm version 2.2.1. The raw data have been deposited in the NCBI Gene Expression Omnibus database (GSE198659).

### CRISPR-mediated gene knockout

VectorBuilder was used to construct and package the lentiviral vectors used for CRISPR-mediated gene knockout in our study: pLV[2CRISPR]-hCas9: T2A: Puro-U6> mPlxna1 [gRNA#1543]-U6> mPlxna1 [gRNA#1169], pLV [2CRISPR]-hCas9: T2A: Puro-U6> mPlxna4 [gRNA#47240]-U6> mPlxna4 [gRNA#46727], and pLV[2CRISPER]-hCas9: T2A: Puro-U6> Scramble [gRNA#1]-U6>. The vector IDs are VB210729-1307uvn, VB211118-1358vvs, and VB211118-1361msk, and can be used to retrieve detailed information about the vectors on vectorbuilder.com. Sorted lung ILC2s were transduced with viral particles 1 d after stimulation with 10 ng/ml rmIL-2 and 10 ng/ml rmIL-33. After 3 d, the infected cells were selected by puromycin culture.

### Statistical analysis

Results are shown as mean ± SEM. All statistical analysis was performed using GraphPad PRISM 8 (GraphPad Software). Statistical comparisons were performed using the non-paired, two-tailed *t* test for the comparison of two groups. One-way ANOVA with Tukey–Kramer post hoc test was used to compare more than two groups. Results were considered statistically significant at $P < 0.05$.

## Supplementary Information

## Acknowledgements

We thank H Matsushita and M Takabatake for their technical assistance. This work was supported by research grants from the Japan Society for the Promotion of Science (JSPS) KAKENHI (JP20K22900 to Y Nakanishi, JP18H05282 to A Kumanogoh); the Japan Foundation for Applied Enzymology (TMFC) (to Y Nakanishi); the Center of Innovation Program (COISTREAM) from the Ministry of Education, Culture, Sports, Science, and Technology of Japan (MEXT) (to A Kumanogoh); the Japan Agency for Medical Research and Development (AMED)-CREST (15652237, to A Kumanogoh); the Japan Agency for Medical Research and Development (AMED) (J210705582, J200705023, J200705710, J200705049, JP18cm016335, and JP18cm059042 to A Kumanogoh); Kansai Economic Federation (KANKEIREN) (to A Kumanogoh); and Mitsubishi Zaidan (to A Kumanogoh).

## Author Contributions

M Naito: conceptualization, investigation, methodology, and writing—original draft.
Y Nakanishi: funding acquisition and investigation.
Y Motomura: methodology and writing—review and editing.
H Takamatsu: investigation and methodology.
S Koyama: investigation and methodology.
M Nishide: investigation.
Y Naito: investigation.
M Izumi: investigation.
Y Mizuno: investigation.
Y Yamaguchi: investigation.
S Nojima: investigation and methodology.
D Okuzaki: investigation.
A Kumanogoh: conceptualization, supervision, funding acquisition, methodology, and writing—original draft, review, and editing.

### Conflict of Interest Statement

The authors declare that they have no conflict of interest.

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
