## [Reviewer comments · Life Science Alliance]

Life Science Alliance

Semaphorin 6D-expressing mesenchymal cells regulate IL-10 production by ILC2s in the lung

Atsushi Kumanogoh, Maiko Naito, Yoshimitsu Nakanishi, Yasutaka Motomura, Hyota Takamatsu, Shohei Koyama, Masayuki Nishide, Yujiro Naito, Mayuko Izumi, Yumiko Mizuno, Yuta Yamaguchi, Satoshi Nojima, and Daisuke Okuzaki

DOI: <https://doi.org/10.26508/lsa.202201486>

Corresponding author(s): *Atsushi Kumanogoh, Osaka University*

Review Timeline:

Submission Date:	2022-04-15
Editorial Decision:	2022-06-27
Revision Received:	2022-08-08
Editorial Decision:	2022-08-17
Revision Received:	2022-08-18
Accepted:	2022-08-18

Scientific Editor: Novella Guidi

Transaction Report:

June 27, 2022

Re: Life Science Alliance manuscript #LSA-2022-01486-T

Prof. Atsushi Kumanogoh
Osaka University
Respiratory Medicine, Allergy and Rheumatic Diseases
Department of Immunopathology, WPI iFrec, Osaka University, 2-2 Yamadaoka
Suita, Osaka 565-0871
Japan

Dear Dr. Kumanogoh,

Thank you for submitting your manuscript entitled "Semaphorin 6D-expressing mesenchymal cells regulate IL-10 production by group 2 innate lymphoid cells in the lung" to Life Science Alliance. The manuscript was assessed by expert reviewers, whose comments are appended to this letter. We invite you to submit a revised manuscript addressing the Reviewer comments.

Thank you for this interesting contribution to Life Science Alliance. We are looking forward to receiving your revised manuscript.

Sincerely,

B. MANUSCRIPT ORGANIZATION AND FORMATTING:

Reviewer #1 (Comments to the Authors (Required)):

Excellent and elegant work that focuses on ILC2 regulation by neuroimmune Sema6D and demonstrates that Sema6D deficiency suppresses type-2 cytokine production but promotes anti-inflammatory cytokine secretion by ILC2s. Sema6D enhances the regulatory function of ILC2s by inducing IL-10 production and these IL-10-producing ILC2 inhibit type 2 cytokines. Defined Sema6D effects on ILC2 are mediated by Plexin A1 receptor. Moreover, Sema6D is an important factor for maintenance of ILC2 numbers in peripheral tissues. This study also demonstrates a critical role for Sema6D in type 2 lung inflammation induced by IL-33 instillation or by *Alternaria alternata*. Noted inconsistencies in the manuscript text: 1. Sema6D enhances the regulatory function of ILC2, and 2. Sema6D suppresses ILC2 regulatory functions. Please correct.

Reviewer #2 (Comments to the Authors (Required)):

LSA-2022-01486-T

Title: Semaphorin 6D-expressing mesenchymal cells regulate IL-10 production by group 2 innate lymphoid cells in the lung

Authors: Atsushi Kumanogoh, Maiko Naito, Yoshimitsu Nakanishi, Yasutaka Motomura, Hyota Takamatsu, Shohei Koyama, Masayuki Nishide, Yujiro Naito, Mayuko Izumi, Yumiko Mizuno, Yuta Yamaguchi, Satoshi Nojima, and Daisuke Okuzaki

Reviewer comments:

Inflammation has been increasingly recognized as an important contributor to pathology in diseases affecting a wide variety of tissues and organs including atherosclerosis, pulmonary hypertension, rheumatoid arthritis, ulcerative colitis, osteoporosis, diabetes, cardiovascular disease and, most recently, neurodegenerative diseases. Molecular mechanisms of local regulation of inflammation are, therefore, of great interest. Prof Atsushi Kumanogoh has pioneered investigation of the role of semaphorins in regulating tissue specific responses. Among other contributions, he and colleagues have described the role of Sema3a in osteogenesis, Sema4A in antibody associated vasculitis, Sema4D in tumor immunity and synovial inflammation, and Sema7a in colitis. The biology of semaphorins is complicated by their variety, the number of sometimes overlapping receptors, and the potential for semaphorins to act either as a ligand or as a reverse signaling receptor themselves. This complexity also lends itself to different patterns of tissue specificity. The present study focuses on Sema6D which can act as either a ligand for plexin-A1 or plexin-A4 receptors or serve as a reverse signaling receptor itself. The authors demonstrate that Sema6D expression promotes IL-5 and IL-13 secretion but suppresses secretion of the anti-inflammatory cytokine IL-10 by activated group 2 innate lymphoid cells (ILC2) of lung. They employ Sema6D deficient (-/-) mice to demonstrate that under conditions of either in vitro or in vivo activation of ILC2, IL-5 and IL-13 mRNA expression and cytokine secretion is reduced while that of IL-10 is increased. This results in reduced local tissue inflammation and reflects altered regulatory activity rather than ILC2 functional exhaustion. Such altered activity is observed in ILC2 isolated from wild type or plexin-A4 but not plexin-A1 knockout mice, indicating that this Sema6D-mediated activity acts through plexin-A1 receptors on ILC2. The highest level of Sema6D expression in lung is seen on mesenchymal cells. It is of interest that, in previous work, Kumanogoh and colleagues reported that Sema6D expression is required to induce anti-inflammatory intestinal macrophage polarization relevant to colitis, and that, in this case, Sema6D mediates reverse signaling induced by plexin-A4 acting as ligand.

The studies reported in this manuscript are well-designed and clearly presented. My only significant criticism is that the authors statement that "Neutralization of IL-10 abrogated the decrease in IL-13 production by Sema6D^{-/-} ILC2s in response to IL-33 and IL-2 (Figure 3I)" (in last paragraph of the Section titled: Sema6D suppresses the regulatory function of ILC2s) is not supported by the data cited in Figure 3I. The informative comparison is between control (no antibody) and addition of anti-IL-10 antibody to cultures of ILC2 derived from lung of Sema6D^{-/-} mice, not between WT and Sema6D^{-/-} illustrated in the figure. In this comparison there is no discernable difference on addition of anti-IL-10. The difference in relative expression of IL-13 transcripts in these same cells shown in Figure 3J, which correctly compares outcome among ILC2 from Sema6D^{-/-} mice, is of modest significance. As a result, the conclusion that IL-10 secretion has a dominant effect on suppressing IL-5, IL-13 and other cytokines is not well-supported.

Comparison of the Sema6D^{-/-} effects on IL-5 and IL-13 vs IL-10 that is shown for in vitro and in vivo stimulation of ILC2s in Figures 2A and 2I indicate that there is greater impact on IL-5 and IL-13 than on IL-10. While there is precedent in the literature

for IL-10 suppression of type 2 cytokines, I am not aware that a more complex chain of events in which suppression of IL-5/IL-13 in Sema6D ^{-/-} enhances IL-10 secretion has been ruled out. This could be addressed by testing antibody neutralization of IL-5 and/or IL-13 as well as IL-10.

The discussion of antibody blockade raises another point. The authors are presumably not insensitive to the translational potential of this work. It would, therefore, be of significant interest to know whether anti-Sema6D antibody could have similar effects on regulation by ILC2s. Anti-Sema6D antibodies have been described, and the availability of Sema6D ^{-/-} should make it relatively straightforward to generate such antibodies (assuming that Sema6D ^{-/-} are not immunocompromised).

These comments do not detract from the overall conclusion that this is solid work from highly qualified investigators that illuminates important and timely issues related to semaphorin activity.

Reviewer #1

*Excellent and elegant work that focuses on ILC2 regulation by neuroimmune Sema6D and demonstrates that Sema6D deficiency suppresses type-2 cytokine production but promotes anti-inflammatory cytokine secretion by ILC2s. Sema6D enhances the regulatory function of ILC2s by inducing IL-10 production and these IL-10-producing ILC2 inhibit type 2 cytokines. Defined Sema6D effects on ILC2 are mediated by Plexin A1 receptor. Moreover, Sema6D is an important factor for maintenance of ILC2 numbers in peripheral tissues. This study also demonstrates a critical role for Sema6D in type 2 lung inflammation induced by IL-33 instillation or by *Alternaria alternata*. Noted inconsistencies in the manuscript text: 1. Sema6D enhances the regulatory function of ILC2, and 2. Sema6D suppresses ILC2 regulatory functions. Please correct.*

→We appreciate the reviewer's favorable comments regarding our study.

In response to the point that there are inconsistencies in the manuscript, we have stated that "... deletion of Sema6D does not induce an exhausted-like phenotype in activated ILC2s, but instead **enhances the regulatory function of ILC2s** by inducing IL-10 production, ..." (Page 11, Lines 8-11), which is consistent with the following context "*In this study, we showed that Sema6D in lung niches suppresses the regulatory functions of ILC2s.*" (Page 14, Lines 2-3). We hope this could be the answer to Reviewer #1 and we would like to appreciate their kind consideration.

Reviewer #2

1) *Inflammation has been increasingly recognized as an important contributor to pathology in diseases affecting a wide variety of tissues and organs including atherosclerosis, pulmonary hypertension, rheumatoid arthritis, ulcerative colitis, osteoporosis, diabetes, cardiovascular disease and, most recently, neurodegenerative diseases. Molecular mechanisms of local regulation of inflammation are, therefore, of great interest. Prof Atsushi Kumanogoh has pioneered investigation of the role of semaphorins in regulating tissue specific responses. Among other contributions, he and colleagues have described the role of Sema3a in osteogenesis, Sema4A in antibody associated vasculitis, Sema4D in tumor immunity and synovial inflammation, and Sema7a in colitis. The biology of semaphorins is complicated by their variety, the number of sometimes overlapping receptors, and the potential for semaphorins to act either as a ligand or as a reverse signaling receptor themselves. This complexity also lends itself to different patterns of tissue specificity. The present study focuses on Sema6D which can act as either a ligand for plexin-A1 or plexin-A4 receptors or serve as a reverse signaling receptor itself. The authors demonstrate that Sema6D expression promotes IL-5 and IL-13 secretion but suppresses secretion of the anti-inflammatory cytokine IL-10 by activated group 2 innate lymphoid cells (ILC2) of lung. They employ Sema6D deficient (-/-) mice to demonstrate that under conditions of either in vitro or in vivo activation of ILC2, IL-5 and IL-13 mRNA expression and cytokine secretion is reduced while that of IL-10 is increased. This results in reduced local tissue inflammation and reflects altered regulatory activity rather than ILC2 functional exhaustion. Such altered activity is observed in ILC2 isolated from wild type or plexin-A4 but not plexin-A1 knockout mice, indicating that this Sema6D-mediated activity acts through plexin-A1 receptors on ILC2. The highest level of Sema6D expression in lung is seen on mesenchymal cells. It is of interest that, in previous work, Kumanogoh and colleagues reported that Sema6D expression is required to induce anti-inflammatory intestinal macrophage polarization relevant to colitis, and that, in this case, Sema6D mediates reverse signaling induced by plexin-A4 acting as ligand.*

→We appreciate the reviewer's sophisticated summary of our previous works on the biological activities of semaphorins. We greatly appreciate the beneficial review of the introduction.

2) *The studies reported in this manuscript are well-designed and clearly presented. My only significant criticism is that the authors statement that "Neutralization of IL-10 abrogated the decrease in IL-13 production by Sema6D-/- ILC2s in response to IL-33 and IL-2 (Figure 3I)" (in last paragraph of the Section titled: Sema6D suppresses the regulatory function of ILC2s) is not supported by the data cited in Figure 3I. The informative comparison is between control (no antibody) and addition of anti-IL-10 antibody to cultures of ILC2 derived from lung of Sema6D-/- mice, not between WT and Sema6D-/- illustrated in the figure. In this comparison there is no discernable difference on addition of anti-IL-10. The difference in relative expression of IL-13 transcripts in these same cells shown in Figure 3J, which correctly compares outcome among ILC2 from Sema6D-/- mice, is of modest significance. As a result, the conclusion that IL-10 secretion has a dominant effect on suppressing IL-5, IL-13 and other cytokines is not well-supported. Comparison of the Sema6D-/- effects on IL-5 and IL-13 vs IL-10 that is shown for in vitro and in vivo stimulation of ILC2s in Figures 2A and 2I indicate that there is greater impact on IL-5 and IL-13 than on IL-10. While there is precedent in the literature for IL-10 suppression of type 2 cytokines, I am not aware that a more complex chain of events in which suppression of IL-5/IL-13 in Sema6D -/- enhances IL-10 secretion has been ruled out. This could be addressed by testing antibody neutralization of IL-5 and/or IL-13 as well as IL-10.*

→We thank the reviewer for their opinion that the studies reported in this manuscript were well-designed and are clearly presented. They did express significant criticism that our data in Figure 3I do not support our statement that "Neutralization of IL-10 abrogated the decrease in IL-13 production by Sema6D-/- ILC2s in response to IL-33 and IL-2 (Figure 3I)" (in the last paragraph of the Section titled: Sema6D suppresses the

regulatory function of ILC2s). The reviewer indicated that the conclusion that IL-10 secretion has a dominant effect on suppressing IL-5, IL-13 and other cytokines is not well-supported.

We agree that the enhanced IL-10 secretion in Sema6D^{-/-} ILC2s does not dominantly contribute to suppressing IL-5/IL-13, and assume that type 2 cytokines in Sema6D^{-/-} ILC2s may be inhibited by other pathways. However, although the effect is limited, the fact remains that the blocking IL-10 increases IL-13 production by Sema6D^{-/-} ILC2s and partially eliminates the difference in IL-13 production between WT and Sema6D^{-/-} ILC2s. Therefore, we toned down our statement and mentioned the possibility that other pathways may suppress type 2 cytokine production in Sema6D^{-/-} ILC2s (Page 11, Lines 2-11).

The reviewer next raised the question of whether Sema6D has a greater impact on IL-5 and IL-13 than IL-10. In addition, they suggested experiments involving antibody neutralization of IL-5 and/or IL-13 as well as IL-10 to exclude the possibility that suppression of IL-5/IL-13 in Sema6D^{-/-} enhances IL-10 secretion.

As suggested, we considered the effects of Sema6D on IL-5 and IL-13 production. When cultured with recombinant Sema6D, WT derived ILC2s stimulated with IL-2 and IL-33 showed decreased IL-10 production. However, IL-5 and IL-13 production was not affected, indicating that cell-extrinsic Sema6D signaling does not directly and dominantly control the production of IL-5 and IL-13. Moreover, ILC2 lack receptors of IL-5 and IL-13, which we confirmed using our RNA-seq data, and this suggest that the reduced production of IL-5 and IL-13 does not affect ILC2 itself. We provided the data below as “Information for the Reviewer #2”.

Information for the Reviewer #2.

Lung ILC2s from WT mice were cultured with Sema6D-Fc or IgG-Fc (10nM each) and stimulated with IL-2 and IL-33 (10 ng/mL each). The amount of IL-10, IL-5 and IL-13 in the supernatant were measured by ELISA.

Normalized FPKM values of the indicated genes in ILC2s from WT mice.

3) The discussion of antibody blockade raises another point. The authors are presumably not insensitive to the translational potential of this work. It would, therefore, be of significant interest to know whether anti-Sema6D antibody could have similar effects on regulation by ILC2s. Anti-Sema6D antibodies have been described, and the availability of Sema6D^{-/-} should make it relatively straightforward to generate such antibodies (assuming that Sema6D^{-/-} are not immunocompromised).

→The reviewer suggested the translational potential of this work, and that it would be of significant interest to know whether anti-Sema6D antibody could have similar effects

on regulation by ILC2s.

According to the reviewer's suggestion, we have discussed the translational potential of Sema6D blockade in the Discussion section of the revised manuscript (Page 14, Lines 19-22). Of note, Sema6D^{-/-} mice are not immunocompromised, and we are now making efforts to generate neutralizing antibody since the quality of commercially available anti-Sema6D antibody is insufficient.

4) These comments do not detract from the overall conclusion that this is solid work from highly qualified investigators that illuminates important and timely issues related to semaphorin activity.

→ We greatly appreciate the reviewer's kind comments.

August 17, 2022

RE: Life Science Alliance Manuscript #LSA-2022-01486-TR

Prof. Atsushi Kumanogoh
Osaka University
Respiratory Medicine, Allergy and Rheumatic Diseases
Department of Immunopathology, WPI iFrec, Osaka University, 2-2 Yamadaoka
Suita, Osaka 565-0871
Japan

Dear Dr. Kumanogoh,

Thank you for submitting your revised manuscript entitled "Semaphorin 6D-expressing mesenchymal cells regulate IL-10 production by ILC2s in the lung". We would be happy to publish your paper in Life Science Alliance pending final revisions necessary to meet our formatting guidelines.

- please upload your manuscript text in an editable doc file format
- please add the Twitter handle of your host institute/organization as well as your own or/and one of the authors in our system
- please add your figure legends to the main manuscript text

To upload the final version of your manuscript, please log in to your account: <https://lsa.msubmit.net/cgi-bin/main.plex>. You will be guided to complete the submission of your revised manuscript and to fill in all necessary information. Please get in touch in case you do not know or remember your login name.

A. FINAL FILES:

B. MANUSCRIPT ORGANIZATION AND FORMATTING:

**Submission of a paper that does not conform to Life Science Alliance guidelines will delay the acceptance of your

manuscript.**

The license to publish form must be signed before your manuscript can be sent to production. A link to the electronic license to publish form will be sent to the corresponding author only. Please take a moment to check your funder requirements.

Sincerely,

Reviewer #1 (Comments to the Authors (Required)):

No additional comments

Reviewer #2 (Comments to the Authors (Required)):

Prof Kumanogoh and colleagues have responded thoughtfully and in full to the main points previously raised by myself and another reviewer. The manuscript is in good order for publication.

August 18, 2022

RE: Life Science Alliance Manuscript #LSA-2022-01486-TRR

Prof. Atsushi Kumanogoh
Osaka University
Department of Respiratory Medicine and Clinical Immunology
2-2 Yamada-oka
Suita, Osaka 565-0871
Japan

Dear Dr. Kumanogoh,

Thank you for submitting your Research Article entitled "Semaphorin 6D-expressing mesenchymal cells regulate IL-10 production by ILC2s in the lung". It is a pleasure to let you know that your manuscript is now accepted for publication in Life Science Alliance. Congratulations on this interesting work.

DISTRIBUTION OF MATERIALS:

Again, congratulations on a very nice paper. I hope you found the review process to be constructive and are pleased with how the manuscript was handled editorially. We look forward to future exciting submissions from your lab.

Sincerely,
